# Entity Disambiguation with Extreme Multi-label Ranking

## ABSTRACT

Entity disambiguation is one of the most important natural language tasks to identify entities behind ambiguous surface mentions within a knowledge base. Although many recent studies apply deep learning to achieve decent results, they need exhausting pre-training and mediocre recall in the retrieval stage. In this paper, we propose a novel framework, eXtreme Multi-label Ranking for Entity Disambiguation (XMRED), to address this challenge. An efficient zero-shot entity retriever with auxiliary data is first pre-trained to recall relevant entities based on linear models. Specifically, the retrieval process can be considered as an extreme multi-label ranking (XMR) task. Entities are first clustered at different scales to form a label tree, thereby learning multi-scale entity retrievers over the label tree with high recall. Moreover, XMRED applies deep cross-encoder as a re-ranker to achieve high precision based on high-quality candidates. Extensive experimental results based on the AIDA-CoNLL benchmark and five zero-shot testing datasets demonstrate that XMRED obtains 98% and over 95% recall scores for in-domain and zero-shot datasets with top-10 retrieved entities. With a deep cross-encoder as the re-ranker, XMRED further outperforms the previous state-of-the-art by 1.74% in In-KB micro-F1 scores on average with a significant improvement on the training efficiency from days to 3.48 hours. In addition, XMRED also beats the state-of-the-art for page-level document retrieval by 2.38% in accuracy and 1.90% in recall@5.

## CCS CONCEPTS

• **Computing methodologies** → **Machine learning**; • **Information systems** → **Information retrieval**.

## KEYWORDS

extreme multi-label classification, entity disambiguation, entity retriever

ACM Reference Format:
Anonymous Author(s). 2024. Entity Disambiguation with Extreme Multi-label Ranking . In *Proceedings of The 2024 ACM Web Conference (WWW '24), May 13–17, 2023, Taipei, Taiwan.* ACM, New York, NY, USA, 8 pages. https://doi.org/10.1145/nnnnnnn.nnnnnnn

## 1 INTRODUCTION

Entity disambiguation is one of the most crucial steps in understanding languages by automatically ironing out references of named entities for various real-world applications, such as entity linking [27], relation extraction [26], and knowledge-aware retrieval [16]. Specifically, entity disambiguation models aim to identify the ground truth entity within a given knowledge base behind its mention, which is a contiguous text span referring to the entity. For example, the mention of "Michael Jordan" could refer to either a professor in a computer science article or a basketball player in sports news, depending on the context.

To capture semantics about the named entity, it is essential to exploit the context information (i.e., the surrounding text of the mention), so contextualized neural language models (NLMs), such as BERT [6] and ELMo [31], have already become a go-to solution in the deep learning era. For instance, NLMs can derive continuous representations of mentions [1, 28] while bi-encoders and cross-encoders can also jointly model candidates for entity classification and ranking [2, 15, 36, 39]. NLM-based sequence-to-sequence models decode entity titles from the mention and its context [3, 5]. However, existing approaches could suffer from their complexity for candidate selection and exhausting pre-training.

A knowledge base is usually enormous with millions of entities. Deep learning models could be too complicated to consider the whole entity space. Accordingly, most of the previous studies rely on a small and pre-defined candidate set derived from the mention-entity prior of a large-scale annotated corpus [10, 30]. However, the dependency on external annotations can be risky and harmful for both accuracy and evaluation. First, the quality of candidate sets significantly affects the task difficulty while different candidate sets result in distinct prediction accuracy for a certain model [40]. Second, the distribution of the prior can be inappropriate to the dataset, especially when mentions may not have annotations in the external corpus [39].

To achieve decent accuracy, training large-scale NLMs for entity disambiguation is challenging because of both data quality and sparsity. As a result, previous approaches usually pre-train their models with external annotations in order to obtain state-of-the-art results. However, pre-training NLMs with an extensive corpus is time-consuming. For example, many studies conducted pre-training with Wikipedia hyperlinks [1–3, 5], which could take weeks even with multiple GPUs. Besides, pre-training on external annotations could also cause information leaks as many benchmark datasets for evaluation are constructed from these signals [12].

In this paper, we propose eXtreme Multi-label Ranking for Entity Disambiguation (XMRED) to address the above challenges. Specifically, we treat entity disambiguation as an eXtreme Multi-label Ranking (XMR) task to retrieve high-quality relevant entities for a mention. First, XMRED derives bag-of-words instance features while a label tree can be built based on positive instance feature aggregation (PIFA) to semantically index all label entities. Second, an XMR-based entity retriever is learned over the label tree so that we can efficiently derive relevant entities from the whole entity space with high recall using beam search. Finally, XMRED learns a BERT-based cross-encoder to re-rank the retrieved entities for achieving better precision on final prediction.

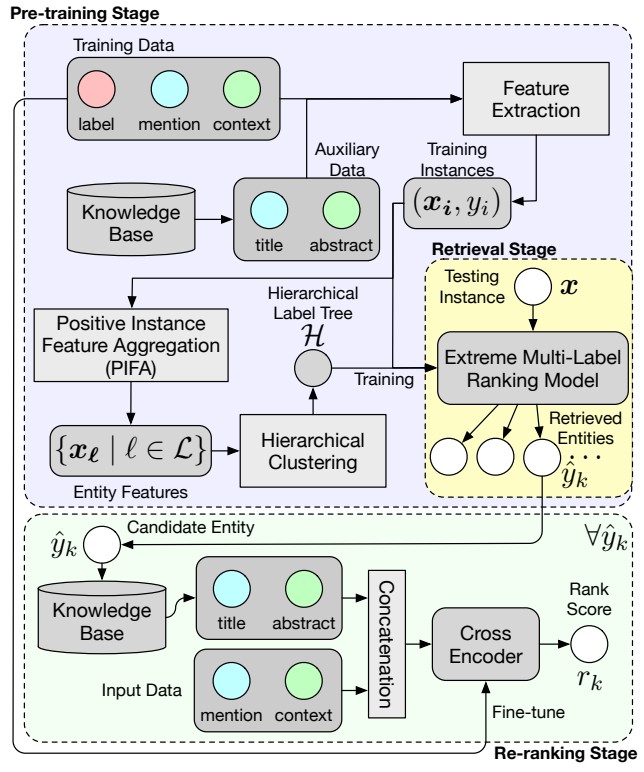

**Figure 1: The overall illustration of the XMRED framework.**

Overall, the contributions of this paper are three-fold.

- First, XMRED establishes the label tree of the entity space so that the hierarchical relations among entities can not only address the data sparsity but also alleviate the need of external annotations for candidate selection.
- Second, we show that simple models and representations, such as linear models with bag-of-words features, are sufficient to retrieve high-quality candidate entities with great efficiency. Specifically, XMRED can obtain 98% and over 95% recall for in-domain and zero-shot datasets.
- Third, we demonstrate that pre-training with a colossal amount of external annotations can be unnecessary when it comes to better candidate entities. Extensive experiments on AIDA-CoNLL and five zero-shot datasets demonstrate that XMRED outperforms competitive baseline methods in entity disambiguation by 1.74% in micro-F1 on average. In addition, XMRED also beats the state-of-the-art results for page-level document retrieval with 2.38% and 1.90% improvements in accuracy and recall@5.

## 2 XMRED: EXTREME MULTI-LABEL RANKING FOR ENTITY DISAMBIGUATION

In this section, we introduce our framework, eXtreme Multi-label Ranking for Entity Disambiguation (XMRED).

**Problem Statement.** Suppose that a knowledge base has a set of entities $\mathcal{L}$ as the space of label entities. Given a document $X =$ $\{x_1, x_2, \ldots, x_{|W|}\}$ that consists of a sequence of tokens $x_i$, for a certain mention $m = [x_{s_m}, \ldots, x_{s_m+|m|}]$, our goal is to identify the ground truth entity behind the mention, where $x_j$ is the $j$-th token of the document; $x_{s_m}$ is the starting token of the mention $m$ within a $|m|$-token span. For simplicity, we define $m$ and its document as an input instance for the machine learning model to determine the entity behind the mention.

**Framework Overview.** Figure 1 shows the illustration of our proposed XMRED framework. XMRED first constructs a hierarchical label tree $\mathcal{H}$ to leverage relations among label entities, thereby training an extreme multi-label ranking model. After retrieving a few relevant entities, XMRED treats them as candidates and learns a deep cross-encoder to provide a rank score $r_k$ for each candidate $\hat{y}_k$.

### 2.1 Bag-of-words Instance Features

XMRED utilizes simple features and models to efficiently retrieve relevant entities for entity disambiguation. In this work, we use unigram and bigram TF-IDF vectors [25] as bag-of-words instance features to represent both mentions and contexts. Formally, we derive the feature vector of a certain mention $m$ by concatenating mention and context features as

$$x_m = [\text{TFIDF}_m(m); \text{TFIDF}_c(X)] \in \mathbb{R}^d, \quad (1)$$

where the functions $\text{TFIDF}_m(\cdot)$ and $\text{TFIDF}_c(\cdot)$ featurize the texts of mentions and contexts into TF-IDF vectors; $d$ is the feature dimension.

### 2.2 Hierarchical Semantic Indexing

With the features of training instances, it is intuitive to learn a machine learning model to compute relevance scores of all label entities for relevance ranking. However, there are two caveats when it comes to entity disambiguation and extreme multi-label ranking. First, the enormous label entity space $\mathcal{L}$ could have millions of entities so that both training and inference would be inefficient. Second, the accuracy for tail entities might fall short because of limited training instances.

To address these issues, we propose to conduct hierarchical semantic indexing for label entities by establishing a label tree based on clustering as shown in Figure 2. Through the label tree, tail entities can leverage other semantically similar entities within the same clusters while the efficiency can be also significantly boosted.

**Positive Instance Feature Aggregation (PIFA).** Following studies in the field of extreme multi-label ranking [44], for entities in $\mathcal{L}$, XMRED adopts positive instance feature aggregation (PIFA) to derive label features that are related to entity disambiguation. Specifically, the label features $z_\ell$ for an entity $\ell \in \mathcal{L}$ can be computed by aggregating the features of mentions whose ground truth entities match the label entity. Given the mention set $\mathcal{M}_\ell$ of the entity $\ell$ in the training data, the PIFA features $z_\ell$ for entity $\ell$ can be computed as follows:

$$z_\ell = \frac{v_\ell}{\|v_\ell\|}, \text{ where } v_\ell = \sum_{m \in \mathcal{M}_\ell} x_m. \quad (2)$$

**Label Tree Construction via Clustering.** To establish a label tree for the entities, XMRED conducts hierarchical clustering. Suppose the root node of a label tree $\mathcal{H}$ represents all of the label entities in

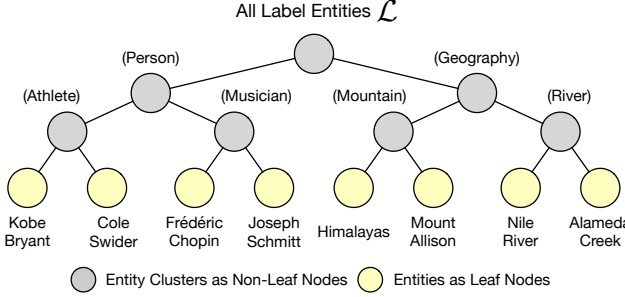

**Figure 2: An example label tree with eight entities. Note that the semantics of nodes are not given, and ideally to be implicitly determined by hierarchical clustering.**

---

**Algorithm 1:** BuildLabelTree($L, Z, K, B$)

**Input:** Entity set $L \subset \mathcal{L}$, label features $Z = \{z_\ell \mid \ell \in L\}$, cluster number $K$, stopping criterion $B$.

**Output:** Constructed label tree $\mathcal{H}$.

1 Let $\mathcal{H} = \{r\}$ with a root $r$ serving all labels;
2 Assign the representative entities $r.L = L$;
3 Initialize the child nodes $\mathcal{N}(r) = \emptyset$;
4 **if** $|L| \le B$ **then**
5    **for** $\ell \in L$ **do**
6       Add a leaf node $u$ into $\mathcal{N}(r)$ into $\mathcal{H}$;
7       $u.L = \{\ell\}$, $u$.label $= \ell$;
8 **else**
9    Perform balanced $K$-means to partition $L$ into $K$ clusters $\{C_i^{(r)}\}$ using $Z$;
10    **for** $i = 1 \ldots K$ **do**
11       $Z_i = \{z_\ell \mid \forall \ell \in C_i\}$;
12       $\mathcal{H}_i = $ BuildLabelTree($C_i^{(r)}, Z_i, K, B$);
13       Add the root of $\mathcal{H}_i$ into $\mathcal{N}(r)$;
14 **return** $\mathcal{H}$;

---

$\mathcal{L}$ while each node exclusively contains a subset of representative entities from its parent. As shown in Algorithm 1, the balanced $K$-Means algorithm recursively partitions the representative entities $L$ of a node $v$ into $K$ clusters $\{C_i\}$ for child nodes $\mathcal{N}(v)$ until $|L|$ meets the stopping criterion $B$. Note that although we adopt balanced $K$-Means as many existing XMC studies [33, 44] for the ease of training and inference, it can simply be replaced with arbitrary clustering algorithms. As a result, $\mathcal{H}$ consists of $O(|\mathcal{L}| \log |\mathcal{L}|)$ nodes, including $|\mathcal{L}|$ leaf nodes for label entities $\ell \in \mathcal{L}$ and other non-leaf nodes that implicitly gather entities with similar semantics.

## 2.3 eXtreme Multi-label Ranking (XMR) for Entity Retrieval

XMRED treats entity retrieval as an XMR task and learns to rank the representative entities $\mathcal{L}_v$ of nodes $v$ in the established label tree $\mathcal{H}$. Given the hierarchical structure of $\mathcal{H}$, XMRED is able to efficiently

perform beam search to identify the most relevant entities for a given input instance.

**One-Versus-All Linear Ranker.** For each non-leaf node $v$ in $\mathcal{H}$, XMRED learns a one-versus-all linear model to rank its child nodes $\mathcal{N}(v)$. Formally, for each child node $u \in \mathcal{N}(v)$, we learn a linear ranker $h_v(x, u)$ parameterized by the model weights $w_{vu} \in \mathbb{R}^d$ as:

$$h_v(x, u) = w_{vu}^\top x. \tag{3}$$

The ranker can then be easily learned by a linear SVM [7] with the following loss function as follows:

$$\sum_{(x,y) \in \mathcal{D}} \sum_{u \in \mathcal{N}(v)} \text{Loss}(x, y, u, h_v) + \frac{\lambda}{2} \sum_{u \in \mathcal{N}(v)} \|w_{vu}\|^2, \tag{4}$$

$$\text{Loss}(x, y, u, h_v) = \max(0, 1 - \mathbb{1}[y \in u.L] \cdot h_v(x, u)), \tag{5}$$

where $\mathcal{D}$ is a proper training dataset; $\mathbb{1}[y \in u.L] \in \{+1, -1\}$ indicates whether the ground truth entity $y$ is covered by the representative entities $u.L$ of the child node $u \in \mathcal{N}(v)$. Finally, we can further obtain a probabilistic rank score for the child $u$ of the node $v$ by applying the sigmoid function as:

$$P(u \mid x, v) = \text{sigmoid}(h_v(x_i, u))). \tag{6}$$

Note that although TF-IDF could result in a high-dimensional space, the features $x$ are actually extremely sparse. Therefore, the ranker is very efficient since the predictions only involve non-zero elements in the feature vector. Moreover, we also prune the model weight $w_{vu}$ by a threshold $\delta$ to further reduce model size and inference cost.

**Hard Negative Sampling.** Using all training instances for training $O(|\mathcal{L}| \log |\mathcal{L}|))$ models is infeasible when the knowledge base has millions of label entities. To tackle this problem, XMRED identifies hard negative samples to not only boost training, but also achieve better performance. Precisely, we utilize Teacher-Forcing Negatives (TFN) [18, 37]. To train the weights $w_{vu}$ for the ranker $h_v$, XMRED collects the eligible TFN samples whose labels are also covered by the parent node $v$ as follows:

$$\{(x_{\text{neg}}, y_{\text{neg}}) \mid y_{\text{neg}} \in u'.L, u' \ne u, u' \in \mathcal{N}(v)\}. \tag{7}$$

**Zero-shot Entity Retriever with Auxiliary Data.** Conventional training data for entity disambiguation is usually sparse. For example, while Wikipedia involves millions of entities, AIDA-CoNLL [14] only contains training instances for thousands of them. To deal with the cold-start issue, XMRED leverages the knowledge base and the metadata from a Cirrus Search Wikipedia dump. Specifically, for each label entity, we treats *title* and *abstract* as the *mention* and *context* to construct a pseudo training instance.

Note that in this paper we do not use the hyperlinks in Wikipedia as additional datasets for learning or pre-training as some previous studies [1, 5, 41]. This is because many benchmark datasets, such as AIDA-CoNLL [14] and WNED-WIKI [11], are actually produced by hyperlinks. Hence, exploiting those signals would lead to leakage and unfair experiments. This phenomenon can also be observed in our study as shown in Table 3, and 4, and 5.

**Fast Inference with Beam Search.** To efficiently retrieve relevant entities, XMRED applies beam search [44] through the label tree as shown in Algorithm 2. Precisely, for each level of the label tree, beam search examines all children $u \in \mathcal{N}(v)$ of searched nodes

**Algorithm 2:** EntityRetriever($\boldsymbol{x}, \mathcal{H}, b, R$)

**Input:** Input features $\boldsymbol{x}$, label tree $\mathcal{H}$, beam size $b$, # of returned entities $R$
**Output:** Relevant entities $[\dots, \hat{y}_k, \dots]$.

1 Let $T$ be the depth of $\mathcal{H}$;
2 Beams = $[\mathcal{H}.\text{root}]$;
3 **for** $t = 2 \dots T$ **do**
4   **if** $|\text{Beams}| > b$ **then**
5     Beams = Beams[:b];
6   Candidates = [];
7   **for** $v \in$ Beams **do**
8     **for** $u \in \mathcal{N}(v)$ **do**
9       Candidates.append($u$);
10   Sort Candidates by the score $p_u$;
11   Beams = Candidates;
12 RelEntities = [];
13 **for** $i = 1 \dots R$ **do**
14   RelEntities.append(Beams[i].label);
15 **return** RelEntities;

| Dataset | Topic | # Docs | # Mentions |
|---------|-------|--------|-----------|
| AIDA (training) | News | 18,448 | 946 |
| AIDA (devlopment) | News | 4,791 | 216 |
| AIDA (testing) | News | 4,485 | 231 |
| MSNBC | News | 20 | 656 |
| AQUAINT | News | 50 | 743 |
| ACE2004 | News | 57 | 259 |
| WNED-CWEB | Web | 320 | 11,154 |
| WNED-WIKI | Wikipedia | 320 | 6,821 |

**Table 1: Statistics of six entity disambiguation datasets.**

| Dataset | # of Mentions | | |
|---------|-------|-----|------|
| | Train | Dev | Test |
| AIDA-YAGO2 | 18,395 | 4,784 | 4,463 |
| WNED-WIKI | N/A | 3,396 | 3,376 |
| WNED-CWEB | N/A | 5,599 | 5,543 |

**Table 2: Statistics of three datasets for the task of page-level document retrieval.**

$v$ from the previous level, and only keeps top-$b$ child nodes in the beam based on their relevance scores $p_u$. When it comes to the bottom level with the leaf nodes, XMRED retrieves the top-$R$ candidates as the retrieved entities. In this study, we leverage the whole search path from the root and estimate the relevance score $p_u$ of a node $u$ as:

$$p_u = p_v \cdot P(u \mid \boldsymbol{x}, v), \tag{8}$$

where $v$ is the parent node of $u$. Therefore, XMRED can obtain top relevant entities in $O(d|\mathcal{L}| \log |\mathcal{L}|)$ computational time complexity. Note that the hyper-parameters $b$ and $R$ are not part of the amortized time complexity because the children of nodes in the same level are mutually exclusive.

### 2.4 Cross-encoder as a Re-ranker

To precisely identify the entity, we further deploy a BERT-based cross-encoder [6] to re-rank the relevant candidates retrieved in XMR. Note that in this study we use the cross-encoder as an example, but the re-ranker can be simply replaced by arbitrary models.

For each retrieved candidate entity $\hat{y}_k$, we concatenate its title and abstract in the knowledge base with the mention and context as the input for the cross-encoder. Specifically, we apply RoBERTa [23] to derive the score $r_k$ for re-ranking as:

$$r_k = \mathcal{F}(\text{RoBERTa}(\text{title}_{\hat{y}_k}</\text{s}>\text{abs}_{\hat{y}_k}</\text{s}>m</\text{s}>c_m)), \tag{9}$$

where $\text{title}_{\hat{y}_k}$ and $\text{abs}_{\hat{y}_k}$ are the title and abstract of the retrieved candidate $\hat{y}_k$ in the knowledge base; $\mathcal{F}$ is a fully-connected hidden layer to produce the ranking score $r_k$. To learn the re-ranker, we simply apply binary cross-entropy [13] as the loss function for optimization. Specifically, for each training mention, we collect top-$R'$ retrieved entities derived by XMRED so that non-hit entities can be considered as hard negative examples. Note that the number of retrieved entities in training $R'$ may differ from the number $R$

used for re-ranking during inference. Finally, the predicted entity is the candidate entity with the highest score $\text{argmax}_{\hat{y}_k} r_k$.

## 3 EXPERIMENTS

In this section, we conduct extensive experiments and in-depth analysis on benchmark datasets to verify the effectiveness and robustness of XMRED in entity disambiguation and page-level document retrieval.

### 3.1 Experimental Datasets

We evaluate XMRED on several benchmark datasets in two tasks: (1) entity disambiguation and (2) page-level document retrieval.

**Entity Disambiguation.** For the task of entity disambiguation, AIDA-CoNLL (AIDA) [14] that retrofits the CoNLL 2003 NER dataset with Wikipedia annotations is considered the benchmark dataset. Specifically, we treat AIDA as the in-domain dataset for training, validation, and testing. Five additional testing datasets, MSNBC, AQUAINT, ACE2004, WNED-CWEB (CWEB) and WNED-WIKI (WIKI) [9, 12], are also included in the experiments as out-of-domain datasets to evaluate the zero-shot capability [1]. Table 1 shows the statistics of the entity disambiguation datasets.

**Page-level Document Retrieval.** For page-level document retrieval, we employ three entity linking datasets in the KILT benchmark [32], including AIDA-YAGO2, WNED-CWEB, and WNED-WIKI. Similar to entity disambiguation, AIDA-YAGO2 serves for training, validation, and testing while WNED-CWEB and WNED-WIKI are two out-domain zero-shot datasets. Note that the labels of testing datasets are not directly provided in KILT while the evaluation process is conducted on the official online evaluation platform. Table 2 shows the statistics of the page-level document retrieval datasets.

| Method | In-Domain | Out-of-Domain (OOD) | | | | | Avg | Avg$_{OOD}$ |
|---|---|---|---|---|---|---|---|---|
| | AIDA | MSNBC | AQUAINT | ACE2004 | CWEB | WIKI | | |
| LNA [10]* | 92.20 | 93.70 | 88.50 | 88.50 | 77.90 | 77.50 | 86.38 | 85.22 |
| RW [12] | 89.00 | 92.00 | 87.00 | 88.00 | 77.00 | 84.50 | 86.25 | 85.70 |
| SGTB [43] | 93.00 | 92.60 | 89.90 | 88.50 | **81.80** | 79.20 | 87.50 | 86.40 |
| LRM [19]* | 93.07 | 93.90 | 88.30 | 89.90 | 77.50 | 78.00 | 86.78 | 85.52 |
| LUD [20]* | 89.66 | 92.20 | 90.70 | 88.10 | 78.20 | 81.70 | 86.75 | 86.18 |
| DCA [42] | 93.73 | 93.80 | 88.25 | 90.14 | 75.59 | 78.84 | 86.73 | 85.32 |
| EntELMo [36] | 93.50 | 92.30 | 90.10 | 88.70 | 78.40 | 79.80 | 87.13 | 85.86 |
| DeepRL [8]* | 94.30 | 92.80 | 87.50 | 91.20 | 78.50 | 82.80 | 87.85 | 86.56 |
| Bootleg [28]* | 80.90 | 80.50 | 74.20 | 83.60 | 70.20 | 76.20 | 77.60 | 76.94 |
| ReFinED [1]* | 93.90 | 94.10 | 90.80 | 90.80 | 79.40 | 87.40 | 89.40 | 88.50 |
| GlobalED[41]* | **95.00** | 94.10 | 91.50 | 90.70 | 78.30 | 87.60 | 89.53 | 88.44 |
| GENRE [5]* | 93.30 | 94.30 | 89.90 | 90.10 | 77.30 | 87.40 | 88.72 | 87.80 |
| GENRE w/o additional annotation | 88.60 | 88.10 | 77.10 | 82.30 | 71.90 | 71.70 | 79.95 | 78.22 |
| ExtEnD [2]* | 92.60 | 94.70 | 91.60 | 91.80 | 77.70 | **88.80** | 89.53 | 88.92 |
| ExtEnD w/o additional annotation | 90.00 | 94.50 | 87.90 | 88.90 | 76.60 | 76.70 | 85.77 | 84.92 |
| XMRED | 94.38 | **95.05** | **92.29** | **97.55** | 81.25 | 87.08 | **91.27** | **90.64** |

**Table 3: In-KB Micro-F1 scores of methods on six entity disambiguation benchmark datasets. (*) denotes the methods that utilize hyperlinks of Wikipedia as additional annotations. Avg and Avg$_{OOD}$ denote the average performance on all six and five out-of-domain datasets. Note that all baseline metrics are reported in their original reports. The best and second-best results are bold and underlined.**

## 3.2 Experimental Setup

**Implementation Details.** We implement XMRED in C++ for fast CPU operations based on PECOS for XMR [44]. TFIDF$_m(\cdot)$ and TFIDF$_c(\cdot)$ treat unigrams and bigrams with top 98% document frequency as the TF-IDF feature spaces for mentions and contexts. To train the label tree $\mathcal{H}$, the cluster number $K$ is set as 16 while the stopping criterion $B$ is 100. The rankers $h_v$ are optimized by the solver of LIBLINEAR [7] with the hyper-parameters $(C, \epsilon, b) = (1, 0.1, 1)$ (i.e., $\lambda = 1$ in Equation 4). The re-ranker of XMRED is built with PyTorch [29] and the Hugging Face Transformer library [38]. We initialize the reranker $\mathcal{F}$ with the SRoBERTa large model [35] as a pre-trained cross-encoder, and then fine-tune it for 10 epochs. AdamW [24] is used for optimization with an initial learning rate 2e-5 and the hyper-parameters $(\beta_1, \beta_2, \lambda) = (0.9, 0.999, 0.1)$. The numbers of retrieved entities $R'$ and $R$ for fine-tuning and inference are set as 15 and 20. All experiments are conducted on an Amazon EC2 p3dn.24xlarge instance with 768 GB memory, 96 CPUs, and 8 NVIDIA Tesla V100 GPUs. All used libraries are feasible for academic research under Apache-2.0 and BSD-3-Clause licenses.

**Comparative Baselines.** The baselines incorporate recent published state-of-the-art works on two benchmarks. RAG [22], T5 [34], and BART [21] are conventional generative models. LNA [10], LRM [19], LUD [20], and Bootleg [28] learn the local attention between context and entities. SGTB [43] conducts structured gradient tree boosting for disambiguation. RW [12] applies random walk algorithms on a mention-entity graph to discover the most relevant entity. DeepRL [8] models the task as a sequence decision problem for reinforcement learning. EntELMo [36], GlobalED [41], and ExtEnD [2] learn discriminate models with pre-trained deep

neural language models. BLINK [39] retrieves candidates with a bi-encoder and re-ranks them with a cross-encoder. GENRE [5], ReFinED [1], and CorpusBrain [3] learn BART-based autoregressive models to decode mentions into entity titles.

## 3.3 Entity Disambiguation Evaluation

Table 3 demonstrates In-KB micro-F1 scores of different methods on six entity disambiguation benchmark datasets. Among the baseline methods, GlobalED, ExtEnD, and ReFinED perform the best against others because they leverage properly pre-trained models and additional annotations from the BLINK data (i.e., 9 million extra training instances). In the task of entity disambiguation, there is no particular edge for either discriminative methods (e.g., GlobalED and ExtEnD) or generative methods (e.g., ReFinED). This could be because the nature of entity disambiguation is actually an (extreme) classification problem. Accordingly, discriminative methods directly learn the classification hypotheses while token prediction in generative methods can also be treated as classifying tokens appeared in entity texts. This finding also demonstrates why learning a high-quality XMR model as a retriever can significantly enhance entity disambiguation.

With the pre-trained XMR model as the retriever and a simple cross-encoder as the re-reranker, XMRED performs the best in both average scores on all six datasets (Avg) and five OOD datasets (Avg$_{OOD}$). An interesting observation is that the performance drops after removing the additional annotations on WIKI are much more significant than the drops on other datasets for both GENRE and ExtEnD. This validates our hypothesis that hyperlinks could result in leakage to some degree as mentioned in Section 2.3. On the other hand, this phenomenon further exhibits the significance and

| Dataset | In-Domain | | Out-of-Domain (OOD) | | | |
|---|---|---|---|---|---|---|
| | AIDA-YAGO2 | | WNED-WIKI | | WNED-CWEB | |
| Method | Dev | Test | Dev | Test | Dev | Test |
| RAG | 77.40 | 72.60 | 49.00 | 48.10 | 46.70 | 47.60 |
| T5 | 86.62 | 74.00 | 47.35 | 47.10 | 46.58 | 49.30 |
| BART | 87.98 | 77.60 | - | 45.90 | - | 49.20 |
| BLINK* | - | 81.50 | - | 80.20 | - | 68.80 |
| GENRE* | 92.75 | 89.85 | 87.69 | 87.44 | 70.57 | 71.22 |
| CorpusBrain* | 92.86 | 89.98 | **88.64** | **88.12** | 71.35 | 70.58 |
| (-add'l annot.) | 90.84 | - | 72.26 | - | 66.23 | - |
| XMRED | **93.96** | **92.36** | 80.12 | 82.32 | **72.21** | **71.95** |
| (Retriever Only) | 85.10 | 79.72 | 76.47 | 76.72 | 67.51 | 67.91 |

**Table 4: Accuracy of methods on the dev and test sets of three page-level document retrieval benchmarks. (\*) denotes the methods that utilize hyperlinks of Wikipedia as additional annotations. Note that all baseline metrics are reported in their original reports while (-) indicates unavailable reports in the original results or the leaderboard. The best and second-best results are bold and underlined.**

robustness of our approach without using Wikipedia hyperlinks as additional annotations.

### 3.4 Page-level Document Retrieval Evaluation

Table 4 shows the accuracy of different methods on the dev and test sets of page-level document retrieval benchmarks. Interestingly, using only the entity retriever (i.e., the pre-trained XMR modle) can beat conventional deep learning models (i.e., RAG, T5, and BART) with only its top-1 prediction. XMRED significantly outperforms all baseline methods in AIDA-YAGO2 and WNED-CWEB. Similar to the situation described in Section 3.3, the performance drop for CorpusBrain discarding additional annotations is also more intense on WNED-WIKI. If we renounce repercussion from the potential leak, XMRED can then beat CorpusBrain without using extra signals.

### 3.5 Retrieval Recall of XMRED Candidates

As an entity retriever, the pre-trained XMR model of XMRED needs to achieve high recall because the recall directly determines the upper-bound of accuracy for re-ranking. Figure 3 illustrates the retrieval performance of XMRED over different numbers of retrieved entities $R$. As a result, XMRED can obtain 96.23% and 98.59% recall scores with only top-5 and top-15 retrieved entities on AIDA for entity disambiguation. For the other five out-of-domain datasets, the recall scores are above 90% when $R$ is greater than 5. For page-level document retrieval, Table 5 shows the recall@5 scores of methods on the dev and test sets of three benchmark datasets. Note that we report recall@5 because the official leaderboard only reports this specific recall position. Similarly, XMRED generally obtains high recall and outperforms deep learning methods on AIDA-YAGO2 and WNED-CWEB. On WNED-WIKI, the recall@5 of XMRED is 91.9% while the two leading baselines are actually in the risk of leakage from their additional Wikipedia annotations. From the above observations, even with only bag-of-words features, XMRED is indeed a strong entity retriever with a great opportunity to supply high-quality candidate entities for downstream re-rankers.

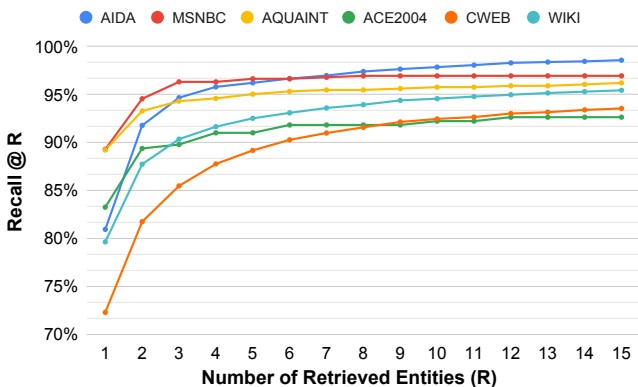

**Figure 3: Retrieval performance of XMRED with top-$R$ retrieved entities on six entity disambiguation datasets.**

| Dataset | In-Domain | | Out-of-Domain (OOD) | | | |
|---|---|---|---|---|---|---|
| | AIDA-YAGO2 | | WNED-WIKI | | WNED-CWEB | |
| Method | Dev | Test | Dev | Test | Dev | Test |
| RAG | 77.4 | 72.6 | 50.0 | 45.2 | 46.7 | 47.6 |
| T5 | 81.8 | 74.1 | 47.4 | 47.1 | 46.6 | 49.3 |
| BART | 86.6 | 77.6 | 47.9 | 45.9 | 48.0 | 49.2 |
| BLINK* | - | 94.8 | - | 91.5 | - | 81.8 |
| GENRE* | - | 94.8 | - | 94.2 | - | 79.2 |
| CorpusBrain* | - | 94.9 | - | **95.6** | - | 78.8 |
| XMRED | **94.7** | **96.7** | **90.6** | 91.9 | **85.2** | **84.7** |

**Table 5: Recall@5 of methods on the dev and test sets of three page-level document retrieval benchmarks. (\*) denotes the methods that utilize hyperlinks of Wikipedia as additional annotations. (-) indicates unavailable reports in the original results or the leaderboard.**

### 3.6 Training Efficiency

The training process of XMRED is efficient. Table 6 shows the training time of different method in pre-training and fine-tuning for AIDA and AIDA-YAGO2. As the pre-training phase, training the XMR model of XMRED only needs 2.25 hours with 96 CPUs. It is indeed significantly faster than other deep learning baselines that require multiple pre-training days with the same hardware resources. For fine-tuning, with a simple cross-encoder structure, XMRED is also more efficient than baselines with complex structures and objectives. Precisely, XMRED is 4.9x/16.4x/5.9x faster than CorpusBrain/BLINK/GlobalED in GPU hours. These results further demonstrate that there is actually no need of significant pre-training to achieve state-of-the-art results when it comes to entity disambiguation. The training time within a day also enables the capability of frequent model refreshment.

### 3.7 Analysis and Discussions

In this section, we have some analysis and discussions.

**Semantics in the Label Tree $\mathcal{H}$.** The label tree $\mathcal{H}$ plays an important role in both training and inference stages of XMRED. Figure 4 depicts part of the constructed label tree. Note that label entities

| Method | Hardware | Pre-training | Fine-tuning |
|---|---|---|---|
| GENRE | 32 GPUs | 22.86 hours | 1.14 hours |
| ReFinED | 4 GPUs | 2 days | 4.32 hours |
| GlobalED | 8 GPUs | 10 days | 7.36 hours |
| BLINK | 8 GPUs | 107.20 hours | 20.18 hours |
| CorpusBrain | 2 GPUs | 3 days | 1 day |
| XMRED | 96 CPUs | 2.25 hours | |
| | 8 GPUs | | 1.23 hours |

**Table 6: Training time of different methods in pre-training and fine-tuning for AIDA and AIDA-YAGO2.**

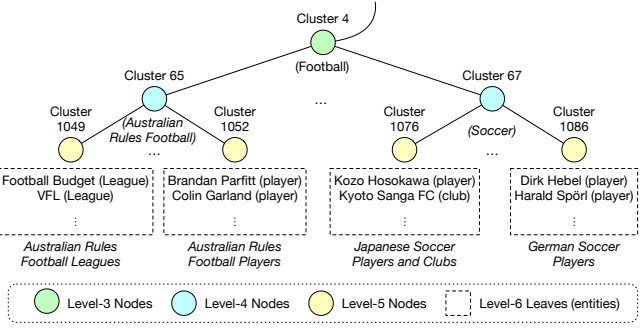

**Figure 4: Illustration of part of the constructed label tree. Note that label entities within dotted boxes are leaf nodes under the corresponding parent level-5 nodes.**

within dotted boxes are leaf nodes under the corresponding parent nodes for simplicity of illustration. First, from the cluster of bottom non-leaf nodes, we can observe that with only instance features, PIFA successfully constructs label features so that semantically similar entities would also share similar features. Second, hierarchical clustering enables multi-scale semantic granularity of clusters over levels of the label tree. The inference process of XMRED can be considered as a reasoning process from broader to narrow semantics.

**Numbers of Retrieved Entities $R$ and $R'$.** The number of retrieved entities is an important hyper-parameter for both fine-tuning and inference. Table 7 presents the In-KB micro-F1 scores of XMRED using different $R$ and $R'$ of retrieved entities for fine-tuning and inference on AIDA. For both $R$ and $R'$, when the numbers of candidates increase from small numbers, the scores would be improved because of more fine-tuning samples and higher recall as shown in Figure 3. However, the performance is dropped with too many retrieved entities. For fine-tuning, it could be because only top-ranked entities are favorable hard negatives. For inference, a longer candidate list can potentially increase the risk of noises when there is only one ground truth for entity disambiguation. According to this study, we set $R'$ and $R$ as 15 and 20.

## 4 RELATED WORK

The recent advances of entity disambiguation are basically a deep learning story. Specifically, many studies adopt pre-trained neural language models to independently [1, 10, 19, 20, 28] or jointly [2,

|  |  | $R$ for Re-ranker Inference | | | | |
|---|---|---|---|---|---|---|
|  |  | 5 | 10 | 15 | 20 | 25 |
| $R'$ for | 5 | 92.32 | 93.50 | 94.01 | 94.15 | 94.10 |
| XMRED | 10 | 92.43 | 93.61 | 94.10 | 94.31 | 94.24 |
| Re-ranker | 15 | 92.50 | 93.68 | 94.22 | **94.38** | **94.38** |
| Fine-tuning | 20 | 91.53 | 92.62 | 93.17 | 93.36 | 93.31 |

**Table 7: In-KB Micro-F1 scores of XMRED using different numbers $R'$ and $R$ of retrieved entities on AIDA.**

36, 41] model mentions and candidate entities with continuous representations to "classify" if the candidate entity is legit for the mention. The other line of research is to treat the task as sequence-to-sequence generation for replacing mentions with entity titles [1, 3, 5, 39]. However, both of these classification and generative approaches are too complicated to appropriately tackle the extreme and sparse entity space of knowledge bases. As a result, they heavily rely on external annotations for pre-training and candidate selection. In contrast, XMRED can efficiently produce high-quality candidates with only bag-of-words features and simply re-rank entities with a simple cross-encoder and fine-tune the model with limited training data.

XMR aims to retrieve a few relevant labels from an enormous space. One line of research is to learn sparse linear models with partitioning techniques, subdividing the label space to smaller spaces for complexity reduction [33, 44]. The other line is to learn latent neural embedding of the input text [4, 17], but neural models with the extreme space usually result in much lower efficiency. In this work, to efficiently retrieve relevant entities, we learn sparse linear models. Besides, to the best of our knowledge, we are the pioneer of using partition-based XMR for entity disambiguation.

## 5 CONCLUSIONS

In this paper, we propose the novel framework, eXtreme Multi-label Ranking for Entity Disambiguation (XMRED), to address the challenges in entity disambiguation. We first show that an extreme multi-label ranking model can be a strong entity retriever for entity disambiguation with only bag-of-words features. The label tree based on positive instance feature aggregation (PIFA) and hierarchical clustering can capture multi-scale semantics of label entities, thereby levering the semantic relations among entities during both training and inference. With a simple cross-encoder as the re-ranker, XMRED can obtain the state-of-the-art performance. In two in-domain and seven out-domain datasets of two benchmarks, XMRED also consistently achieves state-of-the-art performance with not only faster training time, but also exemption from the need of extra annotations.

This work also shows the huge opportunity of mining the nature of entities from their semantic relations. Our analysis indicates the capability of XMRED to automatically compose the semantic structures about entities and their implicit types. The structured semantics has a great potential to further benefit more knowledge-related tasks.

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
