# OpenReview forum: "Entity Disambiguation with Extreme Multi-label Ranking"
_ACM.org/TheWebConf/2024/Conference — TheWebConf24 Oral_

### Official Review · Reviewer_ocLt · 2023-11-06

**Novelty:** 4
**Technical Quality:** 5

**Review:**

The paper proposes XMRED which is a new method for entity disambiguation (linking). XMRED is based on formulating the first retrieval stage of entity linking as an extreme multi-label ranking. The goal is to reduce the huge computational time needed by the contextualized neural language models to achieve a good recall in the first retrieval phase of entity linking. The authors proposed to build a hierarchical label tree for all the entities in the knowledge base using TF-IDF features computed with either Positive Instance Feature Aggregation (PIFA) for mentions of entities in the training dataset, or the title and abstract from the metadata  of the Cirrus Search Wikipedia dump for missing entities in the training dataset. The experimental results show the effectiveness and efficiency of XMRED for the entity disambiguation task.

Pros

S1. The authors proposed to formulate the first retrieval stage of entity linking as a multi-label ranking. This formulation in the entity linking task is novel and interesting as it introduces fast operations compared to the previously proposed contextualized language models for entity linking that requires costly training in the first phase.

S2. Multiple one-versus-all linear SVMs are trained for the internal nodes of the label tree to predict the probabilistic rank score for each node given a TF-IDF representation of an input mention with its context. The computational cost of training is reduced by only considering negative samples from the same parent node.

S3. The authors showed the effectiveness of their proposed method by reporting evaluation metrics for both entity disambiguation and page-level document retrieval, and the efficiency by comparing the training time with multiple baselines for both the pre-training and fine-tuning phases.

Cons

W1. The authors chose to build the label tree using TF-IDF features, but the tree can actually be built from other features such as the text embeddings of mentions and contexts or titles and abstracts. Therefore, the choice of TF-IDF should be experimentally supported.

W2. For the SVM models, as long as the label tree is traversed, the amount of data that is used to train each SVM model is further reduced and this can lead to overfitting of the last SVM models to some specific exact matching tokens.

W3. The pre-trained contextualized language models for entity linking are more suitable than the label tree with SVMs to handle the case of adding a completely new entity to the knowledge base because of the semantic matching signals that are captured by these contextualized language models.

I acknowledge that I have read the rebuttal(s).

**Questions:**

In this paper, the authors proposed a new method, called XMRED, for the entity linking task. The main goal of the paper is to overcome the exhausting pre-trained in the first ranking stage of entity linking. This is achieved by formulating the first ranking stage as an extreme multi-label ranking where the label tree is built from TF-IDF based features, and then efficiently traversed with beam search. There are some points that should be taken into consideration:

1. The mentions and contexts in the training data, and the abstracts and texts in the knowledge base are mapped to TF-IDF vectors. The TF-IDF only captures the exact matching signal, so how does that compare to a text-embedding representation that can capture richer semantics? In addition, the TF-IDF is only computed for unigrams and bigrams, is that enough for a good generalization of the model? What about mentions and contexts that are expressed differently than the pre-computed TF-IDF of the entity? What is the dimension of the TF-IDF vector? Also, applying TF-IDF usually comes with additional text preprocessing to facilitate exact matching, it would be interesting to briefly describe these preprocessing steps in the experimental setup part.

2. As long as the tree is traversed, the number of instances that are used to train an SVM model are reduced. For example, if I understand correctly, for the last level of internal nodes, each node has 100 instances (when setting B to 100 as explained in the experimental setup), so there are 100 SVMs models for this node, and each model is trained with 1 positive instance and 99 negative instances. This may clearly lead to overfitting in the trained SVMs at this level. Please correct me if I’m missing anything, and please comment on this overfitting aspect.

3. In the case of adding completely new entities to the knowledge base, I can see that the pre-trained language models are still suitable given that they are pre-trained to capture rich semantic features. But, in the case of XMRED, how difficult is it to adapt XMRED to newly added entities? Some insights about this aspect can help the reader to have a better understanding about the applicability of the method.

**Reviewer Confidence:**

3: The reviewer is confident but not certain that the evaluation is correct

**Scope:**

4: The work is relevant to the Web and to the track, and is of broad interest to the community

---

### Official Review · Reviewer_4z7X · 2023-11-25

**Novelty:** 4
**Technical Quality:** 3

**Review:**

Summary:
The authors of this paper first design a simple entity retriever to improve its efficiency for pre-training different knowledge bases. Then, the entity retrieval problem is regarded as an eXtreme Multi-label Ranking problem, and the prior probability relied on by previous studies is discarded. Finally, bert is used as the candidate entity and mention reranker.

Strengths:
1. The paper conducts extensive experiments to prove the effectiveness of the method
2. The paper makes a good attempt to abandon the strategy of prior probability, which requires a lot of prior knowledge.

Weaknesses:
1. The framework figure is a little cluttered, and it's hard for me to see what the key point of the framework is.
2. It seems that the newest baseline is in 2022. It would be better to introduce more newest methods in 2023 as baselines.
3. This paper is relatively difficult to understand and requires further effort to refine and polish.

**Questions:**

1. It would be better to provide a clear definition for the eXtreme Multi-label Ranking task, which can be more readable for researchers.
2. I think that "Deep learning models could be too complicated to consider the whole entity space" proposed in the introduction is not very reasonable. Therefore, a simpler model is not necessarily needed as an entity retriever, and for complex entity retrievers the entity representation can be computed in advance without real-time computation.

**Reviewer Confidence:**

3: The reviewer is confident but not certain that the evaluation is correct

**Scope:**

3: The work is somewhat relevant to the Web and to the track, and is of narrow interest to a sub-community

---

### Official Review · Reviewer_A9VL · 2023-11-28

**Novelty:** 5
**Technical Quality:** 6

**Review:**

The paper is well-written
It offers novel entity disambiguation algorithms
Testing with other datasets and methods is thorough

**Questions:**

will XMRED be available on GitHub?

**Ethics Review Description:**

X

**Reviewer Confidence:**

4: The reviewer is certain that the evaluation is correct and very familiar with the relevant literature

**Scope:**

4: The work is relevant to the Web and to the track, and is of broad interest to the community

---

### Official Review · Reviewer_LZmk · 2023-11-29

**Novelty:** 4
**Technical Quality:** 4

**Review:**

This paper proposes an extreme multi-label ranking framework for entity disambiguation. It also applies a deep cross-encoder as a re-ranker given high-quality candidates. Extensive experiments demonstrate both the effectiveness and efficiency of the proposed method, in both in-domain and zero-shot settings.

I have a few questions regarding the methodology design.
1. What if the entities do not follow hierarchical tree structure but instead a graph-like structure with more complex inter-entity relationships?
2. Is there a comparative analysis of deep features versus traditional TF-IDF features within the context of this framework?
3. Is it valid to assume the tree is balanced? How do other clustering algorithms affect the performance and the efficiency of the algorithm?

**Questions:**

Please see my questions in the "Review" part above. Additionally, I am curious about the paper's claim regarding the limitations of sequence-to-sequence approaches in handling sparse entity spaces. Given their ability to utilize semantic relationships among labels, these methods seem well-suited for transferring knowledge from well-represented labels to those with fewer instances.

**Ethics Review Description:**

No need for ethical review.

**Reviewer Confidence:**

2: The reviewer is willing to defend the evaluation, but it is likely that the reviewer did not understand parts of the paper

**Scope:**

3: The work is somewhat relevant to the Web and to the track, and is of narrow interest to a sub-community

---

### Decision · Program_Chairs · 2024-01-22

**Decision:**

Accept (Oral)

**Comment:**

Proposes and evaluates a "an extreme multi-label ranking model" for entity disambiguation. The topic is good fit with the conference. The paper is fun to read. The approach has a high level of novelty. The experiments are thorough and convincing.

 The authors have engaged with the reviewers, and in my opinion, they have adequately addressed all concerns raised by the reviewers. One reviewer, in particular, took time to go back and forth with the authors on several points.

 I note that one review consists of only two lines and gives relatively high scores. Even if I discount this review, I think the remaining reviews provide support for acceptance.

 I think this paper would be a fine candidate for oral presentation since it would attract some interest and attention. I also suspect the authors would give a good presentation.